# Extensive transmission of microbes along the gastrointestinal tract

Thomas SB Schmidt[1†], Matthew R Hayward[1†‡], Luis P Coelho[1§], Simone S Li[1#], Paul I Costea[1], Anita Y Voigt[1¶], Jakob Wirbel[1], Oleksandr M Maistrenko[1], Renato JC Alves[1,2], Emma Bergsten[3], Carine de Beaufort[4,5], Iradj Sobhani[3], Anna Heintz-Buschart[4**], Shinichi Sunagawa[1††], Georg Zeller[1], Paul Wilmes[4], Peer Bork[1,6,7,8*]

[1]Structural and Computational Biology Unit, European Molecular Biology Laboratory, Heidelberg, Germany; [2]Joint PhD programme, European Molecular Biology Laboratory and Faculty of Biosciences, Heidelberg University, Heidelberg, Germany; [3]Department of Gastroenterology and EA7375 -EC2M3, APHP and UPEC Université Paris-Est Créteil, Créteil, France; [4]Luxembourg Centre for Systems Biomedicine, Luxembourg, Luxembourg; [5]Clinique Pédiatrique, Centre Hospitalier de Luxembourg, Luxembourg, Luxembourg; [6]Max Delbrück Centre for Molecular Medicine, Berlin, Germany; [7]Molecular Medicine Partnership Unit (MMPU), European Molecular Biology Laboratory and University Hospital Heidelberg, Heidelberg, Germany; [8]Department of Bioinformatics, Biocenter, University of Würzburg, Würzburg, Germany

*For correspondence:
bork@embl.de

[†]These authors contributed equally to this work

Present address: [‡]The Ragon Institute of MGH, MIT and Harvard, Cambridge, United States; [§]Institute of Science and Technology for Brain-Inspired Intelligence (ISTBI), Fudan University, Shanghai, China; [#]Novo Nordisk Foundation Center for Biosustainability, Technical University of Denmark, Kongens Lyngby, Denmark; [¶]The Jackson Laboratory for Genomic Medicine, Connecticut, United States; [**]Department of Soil Ecology, Helmholtz Centre for Environmental Research - UFZ, Halle, Germany; [††]Department of Biology, ETH Zürich, Zürich, Switzerland

Competing interests: The authors declare that no competing interests exist.

**Abstract** The gastrointestinal tract is abundantly colonized by microbes, yet the translocation of oral species to the intestine is considered a rare aberrant event, and a hallmark of disease. By studying salivary and fecal microbial strain populations of 310 species in 470 individuals from five countries, we found that transmission to, and subsequent colonization of, the large intestine by oral microbes is common and extensive among healthy individuals. We found evidence for a vast majority of oral species to be transferable, with increased levels of transmission in colorectal cancer and rheumatoid arthritis patients and, more generally, for species described as opportunistic pathogens. This establishes the oral cavity as an endogenous reservoir for gut microbial strains, and oral-fecal transmission as an important process that shapes the gastrointestinal microbiome in health and disease.
DOI: https://doi.org/10.7554/eLife.42693.001

## Introduction

Both the oral cavity and large intestine accommodate unique microbiomes that are relevant to human health and disease (*Lynch and Pedersen, 2016*; *Wade, 2013*). Mouth and gut are linked by a constant flow of ingested food and saliva along the gastrointestinal tract (GIT), yet they host distinct microbial communities (*Ding and Schloss, 2014*; *Segata et al., 2012*) in distinct microenvironments (*Savage, 1977*), and have been reported to harbor locally adapted strains (*Lloyd-Price et al., 2017*).

The segregation of oral and intestinal communities is thought to be maintained by various mechanisms, such as gastric acidity (*Howden and Hunt, 1987*; *Martinsen et al., 2005*) and antimicrobial bile acids in the duodenum (*Ridlon et al., 2014*). Failure of this oral-gut barrier has been proposed to lead to intestinal infection (*Martinsen et al., 2005*), and the prolonged usage of proton pump inhibitors can result in an enrichment of particular oral microbes in the gut (*Imhann et al., 2016*).

**eLife digest** Trillions of bacteria and other microbes live in the human body. The mouth and the gut in particular, are microbial hot spots at either end of the digestive tract. Every day, humans swallow around 1.5 liters of saliva, along with millions of oral microbes. Scientists believe that more than 99% of these microbes die as they pass through the acidic environment of the stomach and later the small intestine, which act as a barrier between the bacteria of the mouth and gut.

Failure of this barrier can lead to overgrowth of oral microbes in the gut. This may contribute to diseases like bowel cancer, rheumatoid arthritis and inflammatory bowel diseases. But even in healthy people, low levels of microbes usually found in the mouth are often found in stool. It is unclear if these microbes cross the barrier or if they are similar microbes that originate in the gut.

Now, Schmidt, Hayward et al. show that in healthy people at least one in three oral microbial cells pass through the digestive tract to settle the gut in healthy people. This challenges the notion of a mouth-gut barrier. In the experiments, the genetic material of all the microbes in the saliva and stool of several hundred people from three continents was analyzed. This allowed Schmidt, Hayward et al. to determine whether strains found in the gut originate from the mouth, or are closely related but specialized gut types of the same species. The results also showed that patients with bowel cancer and rheumatoid arthritis had more mouth-to-gut microbial transmission than their healthy counterparts.

The experiments suggest that the mouth is a microbial reservoir that constantly replenishes the gut flora. Some of the gut-traveling oral bacteria trigger inflammation when they grow in other parts of the body like the lining of the heart. This, along with the discovery that patients with certain diseases have more oral bacteria in the gut, may suggest that the transmission of these microbes contributes to disease. The experiments also indicate that finding ways to influence oral bacteria might affect the ones in the gut. More studies are needed to understand how mouth microbes survive the trip to the gut and are able to thrive in this competitive environment, and what role they play in health and disease.

DOI: https://doi.org/10.7554/eLife.42693.002

Increased presence of specific oral taxa in the intestine has in turn been linked to several diseases, including rheumatoid arthritis (*Zhang et al., 2015*), colorectal cancer (*Flynn et al., 2016*; *Zeller et al., 2014*) and inflammatory bowel disease (IBD, (*Gevers et al., 2014*)). While it remains unclear whether disease-associated strains are indeed acquired endogenously (from the oral cavity) or from the environment, it was recently shown that *Klebsiella* strains originating from salivary samples of two IBD patients triggered intestinal inflammation in gnotobiotic mice (*Atarashi et al., 2017*).

This suggests that the presence of oral commensals in the gut is a rare, aberrant event as a consequence of *ectopic* colonization (i.e., 'in the wrong place'), and hence a hallmark of disease. Outside a disease context, however, possible links between the oral and gut microbiome remain poorly characterized. Several genera were shown to be prevalent at both sites (*Segata et al., 2012*), with community types in one being weakly predictive of the other (*Ding and Schloss, 2014*), and with similar gene content in particular species (*Franzosa et al., 2014*), but with distinct, locally adapted strains (*Lloyd-Price et al., 2017*). We hypothesized that this picture is incomplete, and that microbial transmission along the GIT is more common than previously appreciated: that despite oral-gut barrier effects, some microbes freely and frequently traverse the GIT and colonize different niches, forming continuous populations that shape the human microbiome.

## Results and discussion

To test this hypothesis, we assembled and analyzed a dataset of 753 public and 182 newly sequenced saliva and stool metagenomes from 470 healthy and diseased individuals (diagnosed with rheumatoid arthritis, colorectal cancer or type-1 diabetes) from Fiji (*Brito et al., 2016*), China (*Zhang et al., 2015*), Luxembourg (*Heintz-Buschart et al., 2016*), France (*Zeller et al., 2014*), and Germany (*Voigt et al., 2015*) (see Materials and methods, *Figure 1*, and *Supplementary file 1*). For these samples we profiled 310 prevalent species, accounting for 99% of classifiable microbial

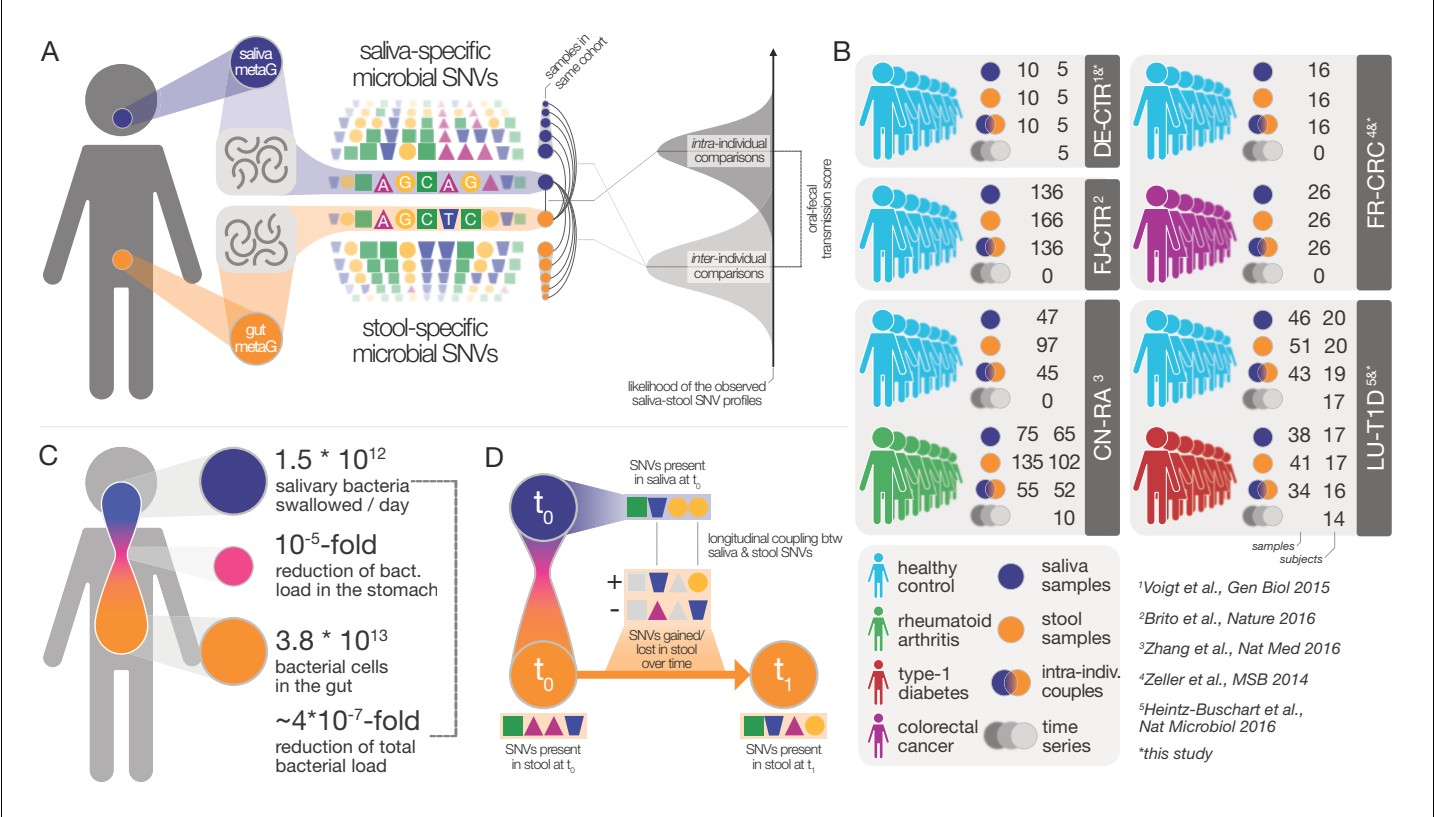

**Figure 1.** Data and workflow overview. (A) Oral-fecal transmission scores were calculated from salivary and fecal microbial SNV profiles. (B) Cohort and dataset overview. For longitudinal cohorts (DE-CTR, CN-RA and LU-T1D), both the total number of samples and the number of individuals are shown, as well as the number of individuals considered in time-series analyses. (C) Salivary and fecal microbial loads allow the calculation of physiologically expected levels of 'passive' microbial transmission (i.e., by ingestion, without growth). (D) The longitudinal coupling of microbial SNVs between salivary and fecal samples was used to infer transmission directionality and oral-fecal transmission rates (see Materials and methods).

DOI: https://doi.org/10.7554/eLife.42693.003

The following figure supplement is available for figure 1:

**Figure supplement 1.** Enrichment of oral species in the gut.

DOI: https://doi.org/10.7554/eLife.42693.004

abundance in both saliva and stool (see Materials and methods and *Supplementary file 2*). We reasoned that if transmission between the oral and gut microenvironments is frequent, we would expect salivary and fecal microbial populations to be more similar within an individual than between individuals. Conversely, under a strong barrier model with restricted transmission, intra- and inter-individual similarities would be equivalent.

We found that at species level, community composition was consistent with distinct populations occupying the oral and intestinal microenvironments. By prevalence across subjects, the 310 profiled species fell into three categories (*Figure 2A*): 44% were predominantly fecal (observed in ≥10% of fecal, but <10% of saliva samples), including core members of the gut microbiome, such as *Clostridium sp.*, *Ruminococcus sp.* and *Bacteroides sp.*; 16% of species were predominantly oral. Although the remaining 125 (40%) species were prevalent in ≥10% of saliva and stool samples, their relative abundances differed greatly between the two habitats. The overall oral and fecal microbiome compositions appeared independent of each other (between-subject Bray-Curtis dissimilarities per site, $\rho_{Pearson}$=-0.03), and the compositional overlap between mouth and gut of the same subject was not found to be significantly different when compared to a between-subject background (Wilcoxon test, Bray-Curtis dissimilarities, p=0.46).

However, to accurately establish and quantify microbial transmission, it is necessary to track populations at the resolution of strains rather than species, as demonstrated previously in fecal microbiota transplantation (*Li et al., 2016*) or seeding of the infant microbiome (*Asnicar et al., 2017*);

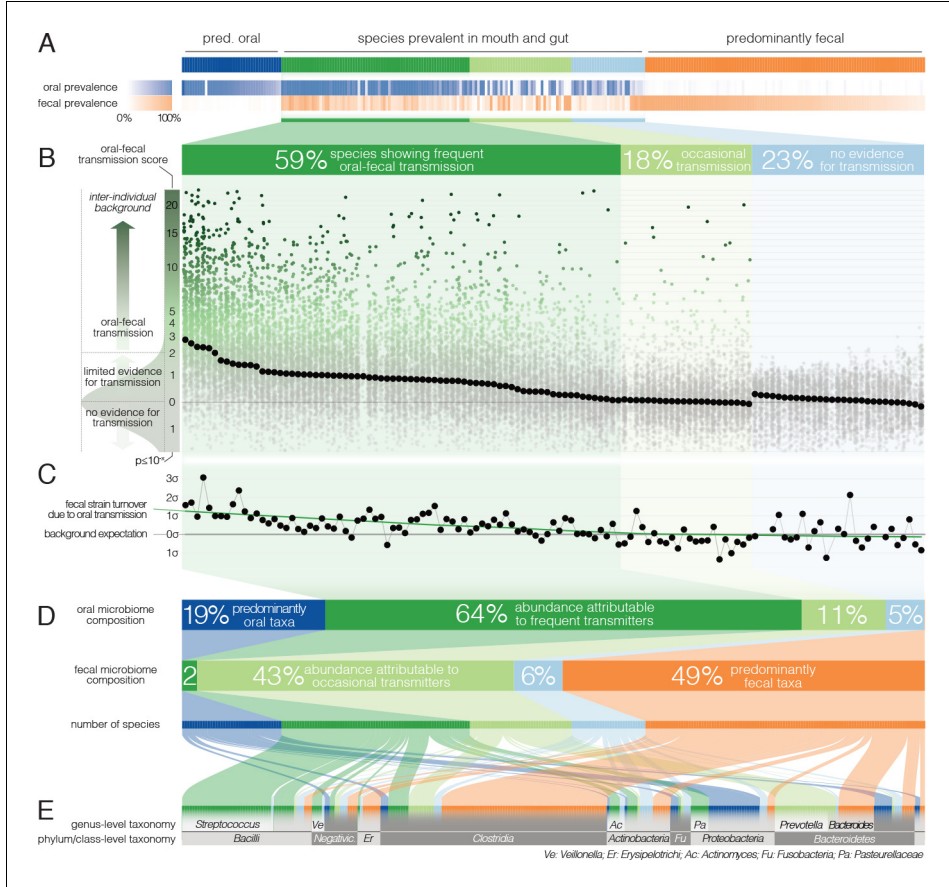

**Figure 2.** Oral-fecal transmission is common across a wide range of phylogenetically diverse species. (**A**) Among 310 tested species, 125 were prevalent in both the mouth and gut across subjects. (**B**) 77% of these formed coherent strain populations between both habitats, when viewed across all tested subjects ('frequent' transmitters) or at least in some ('occasional' transmitters), as evidenced by oral-fecal transmission scores based on intra-individual SNV overlap against an inter-individual background (see Materials and methods). (**C**) Oral-to-fecal transmission rates, as inferred from longitudinal coupling of oral and gut SNVs (see Materials and methods), exceeded background levels for transmitted taxa, even at conservative lower estimates. (**D**) On average, transmissible taxa accounted for a large fraction of classifiable microbial abundance in both the oral cavity and gut. (**E**) Oral-fecal transmissibility was largely a clade-wise trait at genus or family ranks, but common across bacterial phyla.

DOI: https://doi.org/10.7554/eLife.42693.005

The following figure supplements are available for figure 2:

**Figure supplement 1.** Phylogenetic distribution of oral-fecal transmission.

DOI: https://doi.org/10.7554/eLife.42693.006

**Figure supplement 2.** Oral-fecal transmission scores are independent of technical covariates.

DOI: https://doi.org/10.7554/eLife.42693.007

**Figure supplement 3.** Longitudinal stability of SNV profiles per species in saliva and stool.

DOI: https://doi.org/10.7554/eLife.42693.008

**Figure supplement 4.** Directionality of transmission, as inferred from longitudinal data.

DOI: https://doi.org/10.7554/eLife.42693.009

**Figure supplement 5.** Horizontal (breadth) and vertical (depth) coverage cutoffs.

DOI: https://doi.org/10.7554/eLife.42693.010

*Korpela et al., 2018*). We therefore profiled microbial single nucleotide variants (SNVs) across metagenomes, as a proxy for strain populations (*Li et al., 2016*). We formulated a transmission score for each species per subject, based on the likelihood that the observed intra-individual SNV overlap was generated by an inter-individual background model (see Materials and methods). Of the 125 species

prevalent in both mouth and gut, 77% showed evidence of oral-fecal transmission. Out of these, 74 species (59%) showed significantly higher intra-individual SNV similarity across all subjects compared to cohort-wide background SNV frequencies (Benjamini-Hochberg-corrected Wilcoxon tests on transmission scores, p<0.05, see Materials and methods; *Figure 2B*, *Figure 2—figure supplement 1*, *Supplementary file 2*). This suggests that they form coherent strain populations along the GIT in most subjects, subject to frequent oral-fecal microbial transmission. Strains of *Streptococcus*, *Veillonella*, *Actinomyces* and *Haemophilus*, among other core oral taxa, fell into this category. An additional 22 species (18%) showed evidence of at least occasional transmission, with individually significant oral-fecal SNV overlap in some, but not across all subjects, as did 18 species that were generally prevalent in either the mouth or the gut (but not both). All 21 members of the *Prevotella* genus, an important clade of the gut microbiome, were among these occasionally transmitted species. The remaining 29 (23%) species, which were prevalent in both sites, did not show signs of transmission under the strict thresholds we applied.

The fecal abundance of all species with paired observations exceeded lower-bound physiologically predicted levels (i.e., the detection of salivary bacteria in stool purely as the result of ingestion) by several orders of magnitude, even with conservative estimates (*Figure 1C*, *Figure 1—figure supplement 1*). An average person swallows an estimated $1.5 * 10^{12}$ oral bacteria per day (*Humphrey and Williamson, 2001*; *Sender et al., 2016*). Passage through the stomach reduces the viable bacterial load by 5–6 orders of magnitude (*Giannella et al., 1972*; *Sender et al., 2016*), a reduction that is expected to be mirrored at the DNA level, given that free DNA, released from dead bacterial cells, is degraded within seconds to minutes in saliva, the stomach and the intestine (see for example *Mercer et al., 1999* and *Liu et al., 2015*). Relative to the $\sim 3.8*10^{13}$ bacterial cells in the large intestine, 'passive' transmission without subsequent colonization in the gut would therefore account for a reduction in relative abundance by $\sim 4*10^{-7}$ from saliva to feces (*Figure 1C*). Thus, the observed overlap of microbial SNVs could not be explained by passive translocation, but was indeed caused by active colonization in the gut. Moreover, transmission scores across species and subjects were independent of technical covariates, such as the horizontal or vertical coverage of genome mappings (*Figure 2—figure supplement 2*). Average transmission scores across subjects did not correlate with prevalence in stool across all taxa ($\rho_{Spearman} = 0.05$), whereas an association was evident when considering only transmitters ($\rho = 0.67$). In saliva, prevalence was globally indicative of transmission scores ($\rho = 0.6$), reinforcing the notion that core oral taxa tended to be transmitted. Given the limited microbial read depth of salivary metagenomes (due to high fractions of human DNA), this result also indicates that our estimates of oral-fecal transmissibility were quite conservative, with potentially high rates of false negatives.

It was recently shown that during early life, infants are colonized by maternal strains from both the oral cavity and gut (*Ferretti et al., 2018*), and that strains from the latter can persist in the infant gut at least into childhood (*Korpela et al., 2018*). Therefore, to determine whether the observed intra-individual overlap of selected strain populations was due to continuous oral-gut transmission or rare colonization events with subsequent independent expansion in each site, we focused on a subset of 46 individuals for whom longitudinal data was available (with sampling intervals ranging from 1 week to >1 year; mean 79 days). We found that both oral and fecal strain populations were usually stable, even over extended periods of time (*Figure 2—figure supplement 3*), in line with earlier observations for each individual body site (*Lloyd-Price et al., 2017*; *Schloissnig et al., 2013*). Oral and fecal longitudinal SNV patterns were coupled for transmitted species (see Materials and methods): oral SNVs observed at an initial time point were significantly enriched among fecal SNVs that were newly gained over time, but generally not vice versa (*Figure 2—figure supplement 4*). Moreover, oral-fecal transmission rates (i.e., the fraction of fecal strain turnover attributable to oral strains; see Materials and methods) significantly exceeded background expectation for frequently transmitted taxa (*Figure 2C*). These findings orthogonally support the oral-gut transmission hypothesis as they strongly suggest that transmission is in the direction of mouth to gut, and not vice versa; and they imply that oral-intestinal transmission is indeed a frequent and continuous process in which oral strain populations constantly re-colonize the gut.

Oral-fecal transmissibility, as a trait, generally aligned with phylogenetic clade boundaries (phylogenetic signal, $\lambda_{Pagel} = 0.76$), although transmitting groups were found across bacterial phyla (*Figure 2DE*, *Figure 2—figure supplement 1*, *Supplementary file 2*). Transmission scores were negatively correlated with genome size ($\rho_{Spearman} = -0.6$), indicating that transmitted species generally

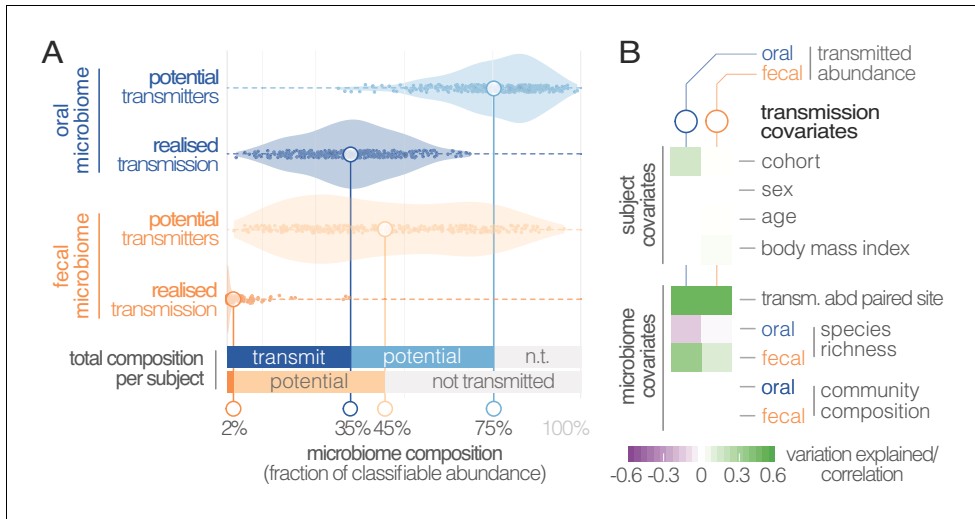

**Figure 3.** Oral-fecal transmission is extensive, with high levels of variation across individuals. (**A**) Potentially transmissible species on average accounted for 75% and 45% of known microbes in salivary and fecal samples, respectively. Among these, realised transmitters were defined as strains that could be traced within subjects with confidence (given detection limits, see Materials and methods). (**B**) Tests for the association of transmission levels in mouth and gut to subject-level covariates (ANOVA, relative sum of squares), to each other ($\rho_{Spearman}$), with oral and fecal community richness ($\rho_{Spearman}$), and with oral and fecal community composition (distance-based redundancy analysis on Bray-Curtis dissimilarities, blocked by cohort, relative sum of squares).
DOI: https://doi.org/10.7554/eLife.42693.011
The following figure supplement is available for figure 3:

**Figure supplement 1.** Multivariable statistical models reveal links between both oral and gut microbiome features with transmission levels.
DOI: https://doi.org/10.7554/eLife.42693.012

had smaller genomes than non-transmitted ones. Moreover, oxygen tolerant species (aerobes and facultative anaerobes) showed 7-fold higher scores than anaerobes on average (ANOVA, p=$10^{-16}$). In contrast, no association was observed for sporulation and motility. To account for possible bias in the species reference and the phylogenetic signal of oral-fecal transmissibility, we confirmed that these signals were robust to phylogenetic regression (*Supplementary file 2*).

Viewed across individuals, we found that seeding of the gut microbiome from the oral cavity was extensive, with high levels of variation (*Figure 3A*). On average, potentially transmissible species (i. e., frequent and occasional transmitters) accounted for 75% of classifiable microbes in saliva, ranging up to 99% in some subjects. However, not all of these were detectable in the matched fecal samples, and oral-fecal strain overlap was generally incomplete. We therefore quantified the fraction of *realized* transmission based on paired observations of species and intra-individual SNV overlap (see Materials and methods). With these criteria, on average 35% of classifiable salivary microbes were transmitted strains that could be traced from mouth to gut within subjects. Similarly, on average 45% (range 2–95%) of classifiable fecal microbes were potential transmitters. These included common fecal species (e.g., *Prevotella copri*) that were detectable in a subset of salivary samples and showed only occasional transmission. Nevertheless, on average only 2% of classifiable fecal microbes could be confidently ascribed to transmitted strains, ranging to >30% in some subjects.

Between-subject variation in the relative abundance of transmitted oral and fecal microbes was found to be independent of subject sex, age and body mass index, although moderate differences were observed between study cohorts (ANOVA, p=0.002; *Figure 3B*; *Supplementary file 3*). Levels of transmitted microbial abundance in mouth and gut were found to correlate with each other ($\rho_{Spearman}$=0.48) and with fecal species richness, but salivary transmitted abundance negatively correlated with oral species richness. This is in line with the observation that core oral species are transmissible, with higher richness implying the increased presence of non-transmitted taxa. Conversely, transmission would add species to a mostly non-transmissible core community in the gut.

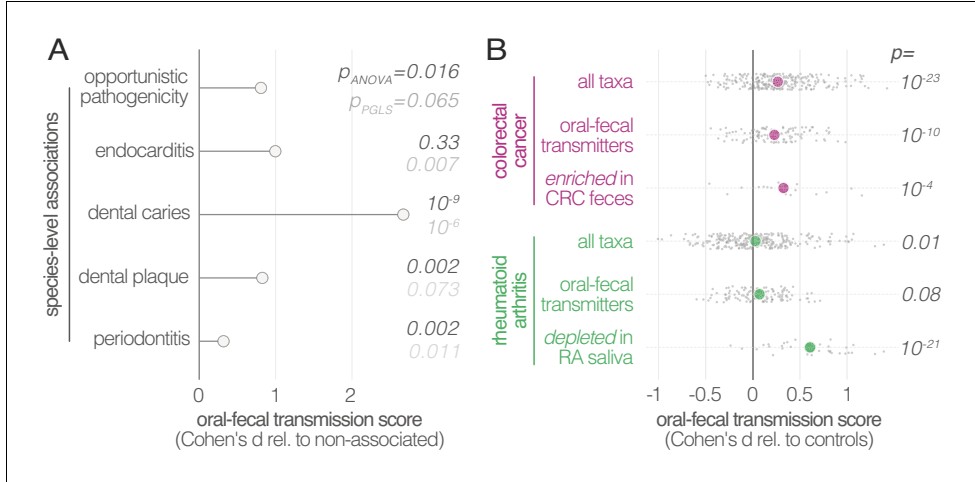

**Figure 4.** Oral-fecal transmission is associated with disease state. (A) Species known to be associated with various diseases showed increased oral-fecal transmission scores ($p_{ANOVA}$, sequential ANOVA including additional phenotypes), even upon phylogenetic generalized least squares regression ($p_{PGLS}$, see Materials and methods and *Supplementary file 2*). (B) Oral-fecal transmission scores tested in colorectal cancer and rheumatoid arthritis cases against controls for specific sets of species (sequential ANOVA, blocked by taxon and subject covariates). Individual data points represent Cohen's d effect sizes (difference in means, normalised by pooled standard deviation) for individual taxa across subjects.

DOI: https://doi.org/10.7554/eLife.42693.013

The following figure supplement is available for figure 4:

**Figure supplement 1.** Species enriched in colorectal cancer show higher oral-fecal transmission scores in patients than controls.

DOI: https://doi.org/10.7554/eLife.42693.014

Although there was no overall association to community composition, levels of transmission correlated with oral or fecal abundances of individual genera (*Supplementary file 3*). To test whether specific oral and gut microbiome features were predictive of transmission, we categorized individuals based on total transmitted abundance in saliva and stool as 'high' or 'low' transmission individuals (Materials and methods). We found that models based on salivary species abundances were mildly predictive of both oral (AUC = 0.738) and fecal (AUC = 0.642) transmission levels (*Supplementary file 4*, *Figure 3—figure supplement 1*). Gut species models, in contrast, were very strong predictors of transmission in both mouth (AUC = 0.951) and gut (AUC = 0.971). This signal was largely driven by the enrichment of transmitting species in stool (*Supplementary file 4*), but surprisingly robust to an elimination of all detected transmitters from the model (AUC = 0.835 for the stool transmission group), again implying that the true extent of oral-intestinal transmission may indeed exceed our conservative estimates. *Fusobacterium nucleatum subsp. animalis* and *nucleatum* stood out among non-trivial gut markers enriched in high-transmission individuals, in line with existing hypotheses that *Fusobacterium nucleatum* subspecies may enable synergistic colonization of oral bacteria in the gut, in association with certain diseases (see for example *Flynn et al., 2016*).

In general, the fecal enrichment of specific oral microbes has repeatedly been associated with various diseases (*Zeller et al., 2014*; *Zhang et al., 2015*). However, due to insufficient taxonomic resolution, oral provenance has so far remained impossible to distinguish from an influx of closely related but distinct strains from the environment. We therefore defined a list of disease states with putative links to oral-fecal transmission and annotated known associations in the literature to all species in our dataset (*Figure 4A*; *Supplementary file 2*). Transmission scores were significantly increased for known opportunistic pathogens (ANOVA, p=0.016), causative agents of dental caries (p=$10^{-9}$), and plaque-dwelling bacteria (p=0.002). Likewise, species associated with periodontitis showed increased evidence for transmission (p=0.002), though this signal was mostly due to mildly periodontic species, while core drivers, such as *Tannerella forsythia*, *Treponema denticola* and *Porphyromonas gingivalis* (*Socransky et al., 1998*), showed little or no indication of oral-fecal transmission.

Endocarditis-associated species showed significantly increased transmission scores upon phylogenetic regression (p=0.007), mostly driven by *Haemophilus*, *Aggregatibacter* and viridans Streptococci. This overall elevated transmissibility of taxa known to colonize ectopically in various habitats across the body (i.e., opportunistic pathogens), in particular via the bloodstream and associated with inflammation (i.e., endocarditis- or periodontitis-associated species (*Hajishengallis, 2015*)), may provide first cues to possible mechanisms of oral-fecal transmission.

Our dataset included metagenomes from case-control studies for rheumatoid arthritis (RA, (*Zhang et al., 2015*)), colorectal cancer (CRC, (*Zeller et al., 2014*)) and type-1 diabetes (T1D, (*Heintz-Buschart et al., 2016*)), totaling 299 individuals, including 172 with salivary and fecal samples. Treatment-naïve CRC patients, sampled before colonoscopy, showed increased transmission scores across all taxa (average per-taxon Cohen's d = 0.27; ANOVA p=$10^{-23}$; *Figure 4B*), as well as for transmitted taxa only (d = 0.23; p=$10^{-10}$). The effect was even more pronounced for species previously described (*Zeller et al., 2014*) to be enriched in the feces of CRC patients (d = 0.33; p=$10^{-4}$; *Figure 4—figure supplement 1*), including *Fusobacterium nucleatum* spp., *Parvimonas micra* and *Peptostreptococcus stomatis*. These findings are in line with a recent report that the oral and fecal microbiome are linked in the context of CRC (*Felmer et al., 2018*), and support the hypothesis (*Flynn et al., 2016*) that CRC-associated species are sourced intra-individually from the oral cavity.

Treatment-naïve RA patients displayed mildly elevated transmissibility across all taxa (d = 0.03, p=0.01) and transmissible taxa only (d = 0.07, p=0.08). Interestingly, species that were orally depleted in RA patients showed markedly increased transmission scores (d = 0.61; p=$10^{-21}$). In contrast, a trend towards decreased transmission in T1D patients was not statistically significant.

Our results demonstrate that influx of oral strains from phylogenetically diverse microbial taxa into the gut microbiome is extensive in healthy individuals, with a high degree of variation between subjects. We showed that the vast majority of species prevalent in both the oral cavity and gut form connected strain populations along the gastrointestinal tract. Furthermore, by leveraging longitudinal data, we established that transmission from the mouth to the gut is a constant process. Approximately one in three classifiable salivary microbial cells colonize in the gut, accounting for at least 2% of the classifiable microbial abundance in feces. This puts oral-fecal transmission well in the range of other factors that determine human gut microbiome composition (*Schmidt et al., 2018*). Moreover, we note that by using saliva and feces as metagenomic readouts, we may underestimate colonization by oral microbes of the mucosa, given that fecal microbiome composition is not fully representative of the gastrointestinal tract (see for example *Zmora et al., 2018*). Therefore, and considering that our estimates of both the number of transmissible species and of the fraction of transmissible microbial abundance are conservative lower bounds due to strict thresholding and current detection limits of metagenomic sequencing, we posit that true levels of transmission are likely even higher, and that virtually all known oral species can translocate to the intestine at least under some circumstances.

Finally, we found increased transmission linked to some diseases, and showed for colorectal cancer and rheumatoid arthritis that disease-associated strains of several species enriched in the intestine are indeed sourced endogenously, that is from the patient's oral cavity, and not from the environment. These results may extend to other diseases beyond those tested here, calling for revised models of microbiome-disease associations that consider the gastrointestinal microbiome as a whole rather than a sum of parts, with important implications for disease prevention, diagnosis, and (microbiome-modulating or -modulated) therapy.

While our findings are observational and do not reveal oral-intestinal transmission routes or mechanistic insights, they challenge current ecological and physiological models of the gastrointestinal tract that assume the oral cavity and large intestine to harbour mostly independent and segregated microbial communities. Instead, most strain populations appear to be continuous along the gastrointestinal tract, originating from the oral cavity, an underappreciated reservoir for the gut microbiome in health and disease.

## Materials and methods

### Metagenomic datasets

Publicly available raw sequence data was downloaded from the European Nucleotide Archive (ENA) for the FJ-CTR (FijiCOMP, project accession PRJNA217052) (*Brito et al., 2016*) and CN-RA (PRJEB6997) (*Zhang et al., 2015*) cohorts. Sample metadata was parsed from ENA and the respective study publications.

For the LU-T1D (PRJNA289586) (*Heintz-Buschart et al., 2016*) cohort, newly generated salivary and fecal metagenomes were added under the existing project accession. For the FR-CRC (ERP005534) (*Zeller et al., 2014*) and DE-CTR (ERP009422) (*Voigt et al., 2015*) cohorts, newly generated metagenomes were uploaded under project accession PRJEB28422 (samples ERS2692266-ERS2692323).

### Sample collection

*German healthy controls (DE-CTR).* Salivary samples were collected at home before dental hygiene and breakfast in the early morning. Donors collected 2–3 ml of saliva and immediately mixed with 15 ml of RNAlater (Sigma-Aldrich). Samples were transported to the laboratory on ice or dry ice and stored at −80C until further processing.

*French colorectal cancer cohort (FR-CRC).* Subject recruitment and cohort characteristics were described previously (*Zeller et al., 2014*). Saliva samples were collected in 1.5 ml saline and stored at −80C until further processing.

*Luxembourg type-1 diabetes cohort (LU-T1D).* Donors collected 2–3 ml of saliva at home before dental hygiene and breakfast in the early morning. Samples were immediately frozen on dry ice, transported to the laboratory and stored at −80C until further processing.

### DNA extraction

*DE-CTR and FR-CRC.* After thawing on ice, 1–2 ml of each sample were centrifuged directly (FR-CRC) or after dilution in RNALater (DE-CTR). Cell pellets were washed 3x in sterile Dulbecco's PBS (PAA Laboratories) and DNA was extracted using the using the GNOME DNA Isolation Kit (MP Biomedicals). Briefly, cell pellets were lysed using a multi-step process of chemical cell lysis/denaturation, bead-beating and enzymatic digestion as described previously (*Zeller et al., 2014*). DE-CTR samples were processed in duplicates, with one replicate being enriched for microbial DNA using the NEBNext Microbiome DNA Enrichment Kit (NEB, Ipswich, USA) following the manufacturer's instructions.

*LU-T1D.* After thawing on ice, two 500 µl aliquots of each sample were centrifuged. Cell pellets were frozen in liquid nitrogen and lysed by cryo-milling and chemical lysis in RLT buffer (QIAGEN). Cell debris was passed through QiaShredder columns (QIAGEN), before DNA was isolated using the QIAGEN AllPrep kit according to the manufacturer's instructions, as described previously (*Heintz-Buschart et al., 2016*).

### Metagenomic sequencing

Libraries for salivary samples of the French and German cohorts were prepared using the NEBNext Ultra DNA Library Prep kit (New England Biolabs, Ipswich) using a dual barcoding system, and sequenced at 125 bp paired-end on an Illumina HiSeq 2000. For the additional LU-T1D samples, libraries were likewise prepared using a dual barcoding system, and sequenced at 150 bp paired-end on Illumina HiSeq 4000 and Illumina NextSeq 500 machines.

### Metagenomic sequence processing

Raw reads were quality trimmed and filtered against the human genome issue 19 to exclude host sequences using MOCAT2, as described previously (*Kultima et al., 2016*). For taxonomic profiling, reads were mapped against a database of 10 universal marker genes for 1753 species-level genome clusters (*specI clusters*, (*Mende et al., 2013*)), using NGless (*Coelho et al., 2018*). A maximum likelihood-approximate phylogenetic tree (with the JTT model, (*Jones et al., 1992*)) for representative genomes of the same 1753 clusters was inferred based on protein sequences of 40 near-universal

marker genes (*Mende et al., 2013*) using the ETE3 toolkit (*Huerta-Cepas et al., 2016*), with default parameters for ClustalOmega (*Sievers et al., 2011*) and FastTree2 (*Price et al., 2010*).

Metagenomic reads were mapped at 97% sequence identity (across at least 45nt) against full cluster-representative genomes, using the Burrows-Wheeler Aligner (*Li and Durbin, 2009*), as implemented in NGless. Reads mapping to multiple genomes at ≥97% identity were discarded from the analysis. Average vertical coverage (sequencing depth) and horizontal coverage (breadth) per microbial genome in each sample were quantified using the qaCompute utility in metaSNV (*Costea et al., 2017*).

Two cohorts (CN-RA (*Zhang et al., 2015*) and DE-CTR (*Voigt et al., 2015*)) contained technical replicates for several salivary samples; these were pooled after the read mapping step.

### Taxa filtering and annotation

The dataset was filtered to include taxa satisfying the following criteria in ≥10% of samples (see *Figure 2—figure supplement 5* for details): horizontal coverage (breadth) of ≥0.05; average vertical coverage (depth) ≥0.25; specI cluster relative abundance of ≥$10^{-6}$. These criteria excluded taxa representing 0.8 ± 1.2% of gut and 1.2 ± 1.9% of oral total mapped abundance. For the remaining 310 taxa, general phenotypes (Gram stain, sporulation, motility, oxygen requirement, among others) were annotated using the *PATRIC* database (accessed Dec 2015) (*Wattam et al., 2017*), and missing values were amended manually. Host and disease association phenotypes (including opportunistic pathogenicity and periodontitis association) were annotated manually, based on published literature and the *MicrobeWiki* website (https://microbewiki.kenyon.edu/index.php/MicrobeWiki, accessed June 2017).

Per taxon summary statistics and annotated metadata are available from *Supplementary file 2*.

### Identification of microbial Single Nucleotide Variants

Microbial Single Nucleotide Variants (SNVs) were called using *metaSNV* (*Costea et al., 2017*). Each potential SNV required support by at least two non-reference sequencing reads (relative to the *specI* cluster representative genomes (*Mende et al., 2013*)) at a base call quality of Phred ≥ 20. The resulting sets of raw SNVs per taxon were filtered differentially for the various downstream analyses, as detailed below.

### Detection of Intra-Individual microbial transmission

To distinguish intra-individual microbial transmission from random drift, we calculated a *transmission score* ($S_T$) per subject and microbial taxon. In short, $S_T$ quantifies how much the similarity between oral and gut SNV profiles *within* an individual deviates from an *inter*-individual background. To calculate $S_T$, we first filtered the set of informative SNVs (all SNVs at a given genome position) by applying the following criteria: (i) observation (read coverage ≥1) at focal position in ≥10 oral and ≥10 gut samples; (ii) SNV observation in ≥1 oral and ≥1 gut sample. Next, we calculated the global background incidence of each allele across oral ($f_{oral}$) and gut ($f_{gut}$) samples. From these, we calculated the background probabilities for each of the four possible cases in paired oral and gut observations: any given allele $i$ could either be present in both samples ($p_{1,1}$), absent in both samples ($p_{0,0}$), or present in one but absent in the other sample ($p_{1,0}$ and $p_{0,1}$):

$$p_{1,1}(i) = f_{oral}(i) * f_{gut}(i)$$

$$p_{0,1}(i) = (1 - f_{oral}(i)) * (1 - f_{gut}(i))$$

$$p_{1,0}(i) = f_{oral}(i) * (1 - f_{gut}(i))$$

$$p_{0,1}(i) = (1 - f_{oral}(i)) * f_{gut}(i)$$

For every permuted oral-gut pair of samples, we then calculated the raw summed log-likelihood of the observed SNV profile overlap ($L_{obs}$) across all alleles with shared coverage:

$$L_{obs} = \left( \sum_{i}^{1,1} log\left(p_{1,1}(i)\right) + \sum_{j}^{0,0} log\left(p_{0,0}(j)\right) \right) - \left( \sum_{k}^{1,0} log\left(p_{1,0}(k)\right) + \sum_{l}^{0,1} log\left(p_{0,1}(l)\right) \right)$$

In other words, $L_{obs}$ quantifies how likely the observed average allele profile agreement between two samples is, given the respective background allele incidence frequencies. Similarly, we computed the log-likelihood of the least likely agreement case ($L_{min}$) per allele:

$$L_{min} = \sum_{i} min\left(log\left(p_{1,1}(i)\right), log\left(p_{0,0}(i)\right)\right)$$

From these values, we calculated a raw probability score ($P_{raw}$) for the observed allele agreement between a given pair of oral and gut samples:

$$p_{raw} = L_{obs}/L_{min}$$

$P_{raw}$ scales the likelihood of the observed agreement by the likelihood of the theoretically most extreme cases of agreement across all observed alleles. In particular, the shared observation of very rare alleles (very low $f_{oral}$ and $f_{gut}$) has a strong impact on $P_{raw}$, whereas the shared observation of very common variants is downweighted.

We computed $P_{raw}$ for all pairwise permutations of oral and gut samples in the dataset with observations (reads) at $\geq 20$ matching positions. We defined the transmission score $S_T(t, s)$ for taxon $t$ in subject $s$ as a standard Z score of the *intra*-individual (within subject) observation against an *inter*-individual (between subjects) background:

$$S_T = (P_{raw}(s) - \mu_{raw})/\sigma_{raw}$$

We tested for potential effects of the choice of background observations by calculating $S_T$ against (i) a global background of all pairwise inter-individual oral-gut comparisons, across all cohorts; (ii) a cohort-specific background per subject; (iii) a global background, but taking only subject-specific comparisons into account (the focal subject's oral sample vs all gut samples, and vice versa); (iv) a within-cohort subject-specific background. Oral-gut comparisons for the same individual across different timepoints, within families (information available for LU and CN cohorts) and within village (for the Fijian cohort) were excluded from the background sets. Although smaller background sets (iii and iv) provided generally noisier scores, overall trends between these backgrounds were very consistent; in particular, cohort-specific vs global backgrounds did not impact trends in our findings (data not shown). All results discussed in the main text therefore refer to scores against a cohort-specific background (ii).

## Quantification of Intra-Individual microbial transmission

To quantify oral-gut transmission per individual, we defined a set of potentially *transmissible* species to include both frequently and occasionally transmitting species. Frequent transmitters encompassed a set of 74 species for which intra-individual transmission scores $S_T$ across subjects were significantly higher than inter-individual background (Benjamini-Hochberg-adjusted one-sided Wilcoxon p<0.05). Occasional transmitters did not satisfy this global criterion, but showed significant evidence for oral-fecal strain overlap in at least one individual (Benjamini-Hochberg-adjusted Z test p<0.05).

To quantify the transmitted microbial abundance per individual, we adjusted the observed relative oral and fecal abundance of each given species by oral-fecal SNV overlap. In other words, the *potentially* transmissible abundance in the oral cavity was defined as the total abundance of potentially transmitting species, and the *realized* transmitted abundance was defined to include only species for which overlapping strain populations could be confidently traced within individuals. This included frequent transmitters that were observable (above detection limits) in matched oral-fecal sample pairs, and occasional transmitters satisfying the additional criterion that significant transmission scores were required in the focal individual for (i.e., an occasional transmitter such as *Prevotella denticola* would only be considered in individuals in which it showed significant transmission scores). For these species, relative oral and fecal abundances were adjusted for total strain population overlap, estimated as the Jaccard overlap of SNVs observed in the oral cavity and gut of the focal individual.

## Longitudinal coupling of oral and fecal SNV profiles

Longitudinal data (2–3 timepoints, see *Supplementary file 1*) was available for 46 individuals from three cohorts (*Heintz-Buschart et al., 2016*; *Voigt et al., 2015*; *Zhang et al., 2015*). To quantify site-specific temporal stability of strain populations, we contrasted within-subject SNV profile similarity over time to between-subject similarities.

Moreover, we tested the longitudinal coupling of strain populations between a putative source site (e.g., oral cavity) and sink site (e.g., gut). For this, we required shared observations (read coverage $\geq 1$) for at least 100 SNV positions across three samples (see *Figure 1*): (i) source site at the initial time point ($t_0$); (ii) sink site at $t_0$; (iii) sink site at a later time point $t_1$. We defined source SNVs as present in sample (i), and newly gained sink SNVs as present in sample (iii) but not (ii), and performed Fisher's exact tests (followed by Benjamini-Hochberg correction) to test for associations between these SNV sets. In other words, we tested for the association of strain populations present in the source site at $t_0$ with strains newly gained in the sink site over time, by proxy of SNV profiles. We considered two sites to be longitudinally coupled in the source - > sink direction if the tested odds ratio was >1 at a (corrected) $p \leq 0.05$. Significant odds ratios < 1 indicated unconnected sites in the tested directionality. Tests were performed independently for oral-to-gut (oral as source, gut as sink) and gut-to-oral coupling, per each taxon.

## Quantification of Oral-Fecal transmission rates

Longitudinal data was also leveraged to estimate oral-fecal transmission rates, here defined as the fraction of fecal strain turnover attributable to the corresponding salivary sample. For each subject and taxon, the absolute fecal strain turnover was quantified as described above, as the difference in SNV profiles between fecal samples at $t_0$ and $t_1$ (samples ii and iii in the previous section). Though sampling intervals ranged from 1 week to >1 year, they were relatively consistent within cohorts (see *Supplementary file 1*). Transmission rates were then quantified as the fraction of fecal alleles gained between $t_0$ and $t_1$ that were also observed in the paired oral sample at $t_0$. Arguably, this provides a conservative lower estimate: oral-fecal transmission could account for both newly gained fecal alleles and for the enhanced stability of existing alleles in the fecal strain population due to a constantly exerted dispersal pressure. However, since the latter effect cannot reliably be quantified from sparse longitudinal metagenomic data, the transmission rates reported in the main text only encompass the former (newly gained alleles).

To test whether transmission rates per taxon were statistically significant across subjects, we compared observed rates to two distinct randomized backgrounds: by shuffling fecal samples at $t_1$ within cohorts, subject-specific *longitudinal* background sets on fecal strain turnover were generated; shuffling oral samples at $t_0$ provided subject-specific *coupled* backgrounds. For each taxon and subject, we Z-transformed observed transmission rates against either of these subject-specific backgrounds; the resulting standard scores (in unit standard deviations) are reported in *Figure 2C*.

## Diversity, Community Composition and Statistical Analyses

Per-sample community richness was calculated from the average of 100 rarefactions to normalised marker gene-based abundances of 1000. Between-sample community compositional similarities were computed as Bray-Curtis and TINA indices, as described previously (*Schmidt et al., 2017*). Distance-based Redundancy Analyses to associate community composition to levels of oral-fecal transmission were performed using the R package *vegan* (*Oksanen et al., 2015*).

The association of transmission scores with taxa phenotypes (oxygen requirement, sporulation, etc.) and taxa disease annotations (opportunistic pathogenicity, etc.) were tested using ANOVA of a combined linear model ('naïve' ANOVA in *Supplementary file 2*). To correct for potentially confounding phylogenetic signals of the tested variables, an ANOVA of a phylogenetically regressed model of the same formulation was performed using the R package *caper* (*Orme et al., 2018*).

Associations of total transmitted classifiable abundance in saliva and stool per subject with subject variables (sex, BMI, age) were tested using ANOVAs on linear models blocked by cohort. The association of transmission scores per subject with disease status was tested using ANOVAs per disease cohort, on linear models accounting for taxon baselines, as well as effects of subject sex, BMI and age.

To test for links between microbiome composition and the amount of transmitted abundance in saliva and stool, we trained machine learning models to classify samples into 'high' and 'low' transmission groups. These groups were defined as the top and bottom quartiles of the fraction of transmitted abundance, independently for stool and saliva samples. For model training, relative abundances were log-transformed and standardized as z-scores. In a 10 times-repeated 10-fold cross-validation setting, L1-regularized (LASSO) logistic regression models (*Tibshirani, 1996*) were trained on the training set and then evaluated on the test set within each fold. In a second step, all species defined as frequent transmitters (see Quantification of Intra-Individual Microbial Transmission above) were eliminated as features before preprocessing and training. All steps (data preprocessing, model building, and model evaluation) were performed using the SIAMCAT R package (https://bioconductor.org/packages/SIAMCAT, version 1.1.0; see also *Zeller et al., 2014*).

All statistical analyses were performed in R. Analysis code is available online (see below).

### Data and analysis code availability

All generated raw sequence data has been uploaded to the *European Nucleotide Archive* under the project accessions PRJEB28422 (French CRC, (*Zeller et al., 2014*) and German German healthy controls, (*Voigt et al., 2015*)) and PRJNA289586 (Luxembourg T1D, (*Heintz-Buschart et al., 2016*)). Sample metadata is available from *Supplementary file 1*. Processed data (taxonomic profiles, taxa annotations, etc.) and full analysis code are available via a gitlab repository (https://git.embl.de/tschmidt/oral-fecal-transmission-public; copy archived at https://github.com/elifesciences-publications/oral-fecal-transmission-public-).

## Acknowledgements

The authors would like to thank Sina Klai of the University of Zürich, Switzerland, Johanna M Schmidt and Gereon Rieke of the University of Bonn, Germany, for helpful comments and discussions on this manuscript, in particular regarding the medical relevance of several of the discussed bacterial species. We thank Katri Korpela, Lucas Silva, Thea van Rossum and other members of the Bork lab at EMBL, Germany, for helpful discussions. We thank Anna M Glazek and Yan Ping Yuan for bioinformatics support, Stefanie Kandels-Lewis of the EMBL for support on sample logistics and administration, Rajna Hercog, Jan Provaznik and Vladimir Benes and, in general, the EMBL Genomics Core Facility for sequencing support, and Laura Lebrun of LCSB for support with the biomolecular extraction platform. TSBS, MRH and AHB were supported by a Luxembourg National Research Fund CORE-INTER grant (MicroCancer; CORE/15/BM/10404093). MRH was additionally supported by a Marie Curie Individual Fellowship (661019). TSBS, SSL, OMM, RJA and PB were supported by an European Research Council grant (MicroBioS; ERC-AdG-669830). GZ and PB were supported by the BMBF-funded Heidelberg Center for Human Bioinformatics (HD-HuB) within the German Network for Bioinformatics Infrastructure (de.NBI #031A537B).

## Additional information

### Funding

| Funder | Grant reference number | Author |
|---|---|---|
| Fonds National de la Recherche Luxembourg | CORE/15/BM/10404093 | Thomas SB Schmidt<br>Matthew R Hayward<br>Anna Heintz-Buschart |
| H2020 European Research Council | ERC-AdG-669830 | Thomas SB Schmidt<br>Simone S Li<br>Oleksandr M Maistrenko<br>Renato JC Alves<br>Peer Bork |
| H2020 Marie Skłodowska-Curie Actions | 661019 | Matthew Robert Hayward |
| German Network for Bioinformatics Infrastructure | de.NBI #031A537B | Georg Zeller<br>Peer Bork |

The funders had no role in study design, data collection and interpretation, or the decision to submit the work for publication.

## Author contributions

Thomas SB Schmidt, Conceptualization, Data curation, Software, Formal analysis, Investigation, Visualization, Methodology, Writing—original draft, Writing—review and editing; Matthew R Hayward, Conceptualization, Data curation, Funding acquisition, Investigation, Methodology, Writing—original draft, Writing—review and editing; Luis P Coelho, Conceptualization, Software, Methodology, Writing—review and editing; Simone S Li, Conceptualization, Methodology, Writing—review and editing; Paul I Costea, Conceptualization, Methodology; Anita Y Voigt, Resources, Investigation; Jakob Wirbel, Formal analysis, Visualization, Writing—review and editing; Oleksandr M Maistrenko, Resources, Investigation, Writing—review and editing; Renato JC Alves, Data curation, Investigation; Emma Bergsten, Investigation, Writing—review and editing; Carine de Beaufort, Iradj Sobhani, Resources, Data curation; Anna Heintz-Buschart, Resources, Data curation, Investigation, Writing—review and editing; Shinichi Sunagawa, Conceptualization, Supervision, Project administration, Writing—review and editing; Georg Zeller, Supervision, Funding acquisition, Methodology, Writing—review and editing; Paul Wilmes, Conceptualization, Supervision, Funding acquisition, Project administration, Writing—review and editing; Peer Bork, Conceptualization, Resources, Supervision, Funding acquisition, Methodology, Writing—original draft, Project administration, Writing—review and editing

## Author ORCIDs

Thomas SB Schmidt https://orcid.org/0000-0001-8587-4177
Luis P Coelho https://orcid.org/0000-0002-9280-7885
Simone S Li https://orcid.org/0000-0002-0073-3656
Renato JC Alves http://orcid.org/0000-0002-7212-0234
Carine de Beaufort http://orcid.org/0000-0003-4310-6799
Anna Heintz-Buschart http://orcid.org/0000-0002-9780-1933
Paul Wilmes http://orcid.org/0000-0002-6478-2924
Peer Bork http://orcid.org/0000-0002-2627-833X

## Ethics

Human subjects: Informed consent was obtained from all study subjects for which novel data was generated; see respective previous publications for details (PMID: 27723761; PMID: 25432777; PMID: 25888008).

## Decision letter and Author response

Decision letter https://doi.org/10.7554/eLife.42693.036
Author response https://doi.org/10.7554/eLife.42693.037

## Additional files

### Supplementary files

• Supplementary file 1. Sample and subject metadata. For a subset of individuals in the CN-RA and DE-CTR cohorts, replicates were merged for salivary samples.
DOI: https://doi.org/10.7554/eLife.42693.015

• Supplementary file 2. Taxa data. Taxa metadata, annotated disease associations, and raw data on relative abundances, horizontal and vertical coverage of each taxon across all samples.
DOI: https://doi.org/10.7554/eLife.42693.016

• Supplementary file 3. Transmission covariates.
DOI: https://doi.org/10.7554/eLife.42693.017

• Supplementary file 4. Abundances of oral and fecal marker species are predictive of transmission levels.
DOI: https://doi.org/10.7554/eLife.42693.018

• Transparent reporting form
DOI: https://doi.org/10.7554/eLife.42693.019

## Data availability

Raw sequencing data have been deposited in the European Nucleotide Archive under project accessions PRJNA289586 and PRJEB28422.

The following datasets were generated:

| Author(s) | Year | Dataset title | Dataset URL | Database and Identifier |
|---|---|---|---|---|
| Schmidt TSB, Hayward MR, Coelho LP, Li SS, Costea PI | 2018 | The Salivary Microbiome in Health and Disease | https://www.ebi.ac.uk/ena/data/view/PRJEB28422 | European Nucleotide Archive, PRJEB28422 |
| Schmidt TSB, Hayward MR, Coelho LP, Li SS, Costea PI, Voigt AY, Maistrenko OM, Alves RJ, Bergsten E, de Beaufort C, Sobhani I, Heintz-Buschart A, Sunagawa S, Zeller G, Wilmes P, Bork P | 2018 | Human Gut Microbiome in a Multiplex Family Study of Type 1 Diabetes Mellitus | https://www.ebi.ac.uk/ena/data/view/PRJNA289586 | European Nucleotide Archive, PRJNA289586 |

The following previously published datasets were used:

| Author(s) | Year | Dataset title | Dataset URL | Database and Identifier |
|---|---|---|---|---|
| Zhang X, Zhang D, Jia H, Feng Q, Wang D, Liang D, Wu X, Li J, Tang L, Li Y, Lan Z, Chen B, Zhong H, Xie H, Jie Z, Chen W, Tang S, Xu X, Wang X, Cai X, Liu S, Xia Y, Qiao X, Al-Aama JY, Chen H, Wang L, Wu QJ, Zhang F, Zheng W, Zhang M, Luo G, Xue W, Xiao L, Yin Y, Yang H, Wang J, Kristiansen K, Liu L, Li T | 2015 | The oral and gut microbiomes are perturbed in rheumatoid arthritis and partly normalized after treatment. | https://www.ebi.ac.uk/ena/data/view/PRJEB6997 | European Nucleotide Archive, PRJEB6997 |
| Brito IL, Yilmaz S, Huang K, Xu L, Jupiter SD, Jenkins AP, Naisilisili W, Tamminen M, Smillie CS, Wortman JR, Birren BW, Xavier RJ, Blainey PC, Singh AK, Gevers D | 2016 | FijiCOMP: saliva and stool metagenomes | https://www.ebi.ac.uk/ena/data/view/PRJNA217052 | European Nucleotide Archive, PRJNA217052 |
| Voigt AY, Costea PI, Kultima JR, Li SS, Zeller G, Sunagawa S | 2015 | Temporal and technical variability of human gut metagenomes. | https://www.ebi.ac.uk/ena/data/view/PRJEB8347 | European Nucleotide Archive, PRJEB8347 |
| Zeller G, Tap J, Voigt AY, Sunagawa S, Kultima JR, Costea PI, Amiot A, Böhm J, Brunetti F, Habermann N, Hercog R, Koch M, | 2014 | Potential of fecal microbiota for early-stage detection of colorectal cancer. | https://www.ebi.ac.uk/ena/data/view/PRJEB6070 | European Nucleotide Archive, PRJEB6070 |

| Luciani A, Mende DR, Schneider MA, Schrotz-King P, Tournigand C, Tran Van Nhieu J, Yamada T, Zimmermann J, Benes V | | | | |
|---|---|---|---|---|
| Heintz-Buschart A, May P, Laczny CC, Lebrun LA, Bellora C, Krishna A | 2016 | Integrated multi-omics of the human gut microbiome in a case study of familial type 1 diabetes. | https://www.ebi.ac.uk/ena/data/view/PRJNA289586 | European Nucleotide Archive, PRJNA289586 |

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
