## [Decision Letter]

Thank you for submitting your article "Extensive Transmission of Microbes along the Gastrointestinal Tract" for consideration by *eLife*. Your article has been reviewed by Wendy Garrett as the Senior Editor, a Reviewing Editor, and three reviewers. The following individuals involved in review of your submission have agreed to reveal their identity: Andrei Prodan (Reviewer #1); Paul O'Toole (Reviewer #3).

The reviewers have discussed the reviews with one another and although the paper is currently now suitable for publication, the Reviewing Editor has drafted this decision to help you prepare a revised submission.

Summary:

This paper by Schmidt et al., is an overall well written manuscript that shows that bacterial strain exchange between the oral and gut environments is more extensive than previously thought. Additionally, the manuscript puts forth that this is a normal occurrence, rather than solely a mark of dysbiosis / disease and that opportunistic pathogens had higher evidence of transmission along the gastrointestinal tract with an extensive exchange of strains between the two using SNV profiles of each bacterial strain with oral cavity bacterial strain dominance.

Essential revisions:

There are a few methodological questions that needs to be addressed. This includes cross checking with the ConStrains method to identify microbial strains (Luo et al., 2015. Also, the amount of sequence coverage per sample (in Gbp) should be specified so that readers have a platform-independent reference point for the coverage that is necessary and sufficient for SNV analysis. Moreover, as reviewer 3 points out the authors should further expand on their estimations of quantities and viability of bacterial cells in the lumen that can be attributed to salivary bacteria. In this regard, the paper could be strengthened if there would be some data on viability of bacterial strains in the intestinal tract as the authors might have underestimated the contribution of passively translocated DNA (belonging to living or dead bacteria) to the faecal microbiota. In this regard, a few years ago, Korem et al., (2015) published about this using shotgun metagenomic sequencing to calculate the ratio of sequencing coverage between the peak and trough providing a quantitative measure of a species' growth rate. It would be of importance to see if this bioinformatic approach would help solve this question. Finally, a dedicated section on statistical analysis in the method description of the paper would be helpful and the numbers of included individuals should be cross checked within the supplemental data and tables (e.g. Figure 1).

Please see the full reviews below for further points:

Reviewer #1:

This is a solid paper which shows that bacterial strain exchange between the oral and gut environments is more extensive than previously thought and a normal occurrence, rather than solely a mark of dysbiosis / disease. Nonetheless, they also show that opportunistic pathogens had higher evidence of transmission. While the authors find no correlation between the β-diversity measures (as determined by metagenomic profiling) of the gut and the oral environment, they show that there is an extensive exchange of strains between the two. This is done by determining the strain profiles (SNV profiles of each strain) and showing that, based on probabilistic models, the overlap in some of these profiles is significantly higher than would be expected by random chance ("transmission scores"). The fact that species transmission scores correlate with oral relative abundance, but not to gut abundance, indicate that (as would be expected) the direction of transmission is from the oral cavity to the gut as does the fact that 'oral SNVs observed at an initial time point were significantly enriched among fecal SNVs that were newly gained over time, but generally not vice versa '. The paper is made stronger by the use of longitudinal data.

The paper is concise and well-written, providing a detailed, clear and understandable description of the methods (e.g. how the transmission scores are calculated and what the rationale is). The visualizations (including the Supplementary figures) are very expressive and informative. Supporting information files have been submitted, and code and data have been made available on a GitHub repository.

I have very few critical remarks to make, I think the paper is rigorous and well-polished:

1) PRJEB28422 accession number does not exist on ENA (checked on Nov 27th). Did the authors mean PRJEB22368?

2) "Transmission scores were negatively correlated with genome size (ρ Spearman=-0.6), indicating that transmitted species generally had smaller genomes than non-transmitted ones" Any idea why this is the case?

3) "the fecal relative abundance of *Fusobacterium sp.* positively correlated with higher levels of transmission)" Why would this be the case?

Reviewer #2:

In this work, Schmidt et al., provide theoretical evidence for the transmission and colonization of oral microbial genomes in the distal gut. The results are interesting and important because they show that transmission of oral microbes to the gut occurs extensively in healthy individuals also in adulthood. Therefore, if these results are confirmed, oral-gut transmission might be an important factor to consider for the prevention and management of human diseases through the GIT microbiota.

We have a few major concerns that the authors should address to support their findings:

1) How were the cut-offs for vertical and horizontal genome coverage chosen? A 5% breadth of coverage seems low for the identification of microbial strains. Would the choice of these cut-offs affect the results? Indeed, Supplementary figure 4 shows that both vertical and horizontal genome coverage can affect the transmission score at least of some specific taxa. Please highlight the distribution of transmitters in this supplementary figure.

2) The authors base their work on the assumption that oral and gut SNVs profiles of transmitted genomes are more similar in an individual than between individuals. However, does this similarity of SNVs profiles necessarily imply transmission? Or, alternatively, could other individual-specific genetic and/or environmental factors shape a similar oral and gut microbiota in an individual instead of transmission?

3) As the authors use a low breadth of coverage and assume transmission based on similarity of SNVs profiles, we would like to ask that the authors confirm their results when using the ConStrains method to identify microbial strains (Luo et al., 2015). This method is also based on SNPs in oral and gut samples.

Reviewer #3:

This study entitled "Extensive Transmission of Microbes along the Gastrointestinal Tract" is an original work focusing on the transmission of bacterial strains between the oral and gut environment. The study is a robust analysis at strain-level of the oral and gut microbiota composition intra- and inter-individuals. The data are well exploited, especially regarding potential confounding factors between the different cohorts. The authors set out to identify population flow of bacteria from the oral cavity to the lumen. They defined metagenomes to an SNV-level resolution. The central thesis of this paper is that intra-individual overlap of SNVs between the oral cavity and the lumen is greater than that which would be expected from inter individual background thereby demonstrating oral taxa translocation. Within their model this event of translocation was a persistent one. The taxa identified as transmitted were phylogenetically diverse, yet some clade clustering was noted. The only noted characteristics of these taxa were reduced relative genome size and their anaerobic/ facultative aerobic nature.

This is a novel study with potentially significant ramifications for human intestinal microbial ecology.

My main critiques are these four points:

1) One of the pillar arguments in this paper is that the numbers of bacteria cells in the colon which show evidence of transmission, cannot be attributed to the passive translocation due to peristalsis. They argue that the amount of bacterial cells swallowed by a human per day (1.5*1012) would be depleted during passage through the upper digestive tract (stomach and duodenum) and thus would not contribute significantly to the gut microbiota. However, methodologically, the authors are investigating DNA not viable living cells. One might argue the authors have underestimated the contribution of passively translocated DNA (belonging to living or dead bacteria) to the faecal microbiota. The stomach and the mouth contain a high proportion of dead cells (perhaps only 1% of stomach bacteria cells are alive) whose DNA could translocate to the lumen. Given the estimates of an average person has 1 bowel moments a day (the cohort is older with disease so could be less), the average mass of stool to be 100 grams and the bacterial density in stool is estimated to be 0.9·1011 bacteria/g; an individual would pass 9x1012 bacterial cells a day. Presuming that an individual passes all the saliva they swallow a day in their bowel movement and there is no loss of bacterial DNA; One would detect ~1.5*1012 per stool. This would be above the 10% limit that the authors set. I think the authors should further expand on their estimations of quantities of bacterial cells in the lumen that can be attributed to salivary bacteria. The literature to which they reference in not primary work in essence and review. More solid numbers are needed when discussing the oral bacterial density and volume. Indeed, the focus of the Sender et al., 2016 paper is the colonic microbiota. I think their strong claims need stronger support. Likewise, the results presented in the study are not enough to support the following statement in particular "Approximately one in three salivary microbial cells colonise in the gut, accounting for at least 2% of the classifiable microbial abundance in feces".

2) The number of individuals reported in the text does not match the cohort and dataset overview presented in Figure 1. In the abstract and the main text, the authors reported the analysis of 470 healthy and diseased individuals but based on Figure 1 all together the cohorts comprised 571 individuals including 365 intra-individual couples. Further, the authors reported in the main text they they focused on a subset of 57 individuals for whom longitudinal data was available but based on Figure 1 only 46 individuals (including diseased individuals) presented time series. Then, for the case-control studies, the authors reported in the main text a total of 172 individuals but based on Figure 1 the cohorts CN-RA, FR-CRC and LU-T1D comprised 395 individuals including 219 intra-individual couples (healthy and diseased individuals). As a general comment, the authors should also clarify precisely in the main text whether the studied individuals are intra-individual couples (with both saliva and stool samples) or individuals with one sample type.

3) The authors profiled 310 prevalent species, which accounted for 99% of classifiable microbial abundance in both saliva and stool. However, there is no mention of the unclassifiable fraction of the reads, the proportion of classified over non-classified reads or the percentage of mapped reads.

4) The authors should acknowledge the difference between colonization of the lumen (faecal matter) versus colonization of the mucosa. Recent work by Zmora et al., 2018 on probiotics have highlighted the disparity between the colonization of the faecal matter versus the mucosa. I recognize they prepared the current submission before the Zmora paper came out. However, there are many other papers that make this point, at least with respect to faecal versus mucosa.

[Editors' note: further revisions were requested prior to acceptance, as described below.]

Thank you for sending your article entitled "Extensive Transmission of Microbes along the Gastrointestinal Tract" for peer review at *eLife*. Your article is being evaluated by three peer reviewers, and the evaluation is being overseen by a Reviewing Editor and Wendy Garrett as the Senior Editor.

Reviewer #2:

We thank the authors for their work while revising their manuscript. The manuscript reads well, and we have only one additional comment concerning the correlations between average transmission scores and other parameters provided in the supplementary table.

We could reproduce the results (Results and Discussion section) for the correlations between transmission scores and prevalence_saliva (rho=0.6) as well as prevalence_gut (rho=0.05). However, there was a stronger correlation between average transmission score and prevalence_gut when accessing only the transmitters.

Additionally, strong correlations were observed for average transmission score and P/horizontal coverage for transmitters.

Can the authors comment?

---

## [Author Response]

Summary:This paper by Schmidt et al., is an overall well written manuscript that shows that bacterial strain exchange between the oral and gut environments is more extensive than previously thought. Additionally, the manuscript puts forth that this is a normal occurrence, rather than solely a mark of dysbiosis / disease and that opportunistic pathogens had higher evidence of transmission along the gastrointestinal tract with an extensive exchange of strains between the two using SNV profiles of each bacterial strain with oral cavity bacterial strain dominance.

We thank the editor for this positive feedback, and for soliciting constructive reviews that raised relevant and interesting suggestions to further strengthen our study.

Essential revisions:There are a few methodological questions that needs to be addressed. This includes cross checking with the ConStrains method to identify microbial strains (Luo et al., 2015.

Following this suggestion, we ran ConStrains on a subset of 144 samples from three cohorts of our dataset. Although the program finished successfully, it did not detect any strain heterogeneity in any of these samples due to “insufficient coverage” (see results tables attached to this review). The same behavior was previously reported by Quince et al., 2017 even on simulated mock community data:

“We were unable to run ConStrains [15] on the same data set, as the program complained that insufficient coverage of *E. coli* specific genes was obtained from the MetaPhlAn mapping. This is despite the fact that the *E. coli* coverage across our samples ranged between 37.88 and 432.00, with a median coverage of 244.00, well above the minimum of 10.0 stated to be necessary to run the ConStrains algorithm.”

Moreover, we have several theoretical arguments why tools like ConStrains are less powered to pick up the observed signal of oral-fecal strain overlap, as detailed below in the response to the original comment by reviewer #2.

Also, the amount of sequence coverage per sample (in Gbp) should be specified so that readers have a platform-independent reference point for the coverage that is necessary and sufficient for SNV analysis.

We have now added the total depth of microbial reads per sample to supplementary file 1, and the horizontal (breadth) and vertical (depth) coverage for each species in each sample as additional sheets to Supplementary file 2.

Moreover, as reviewer 3 points out the authors should further expand on their estimations of quantities and viability of bacterial cells in the lumen that can be attributed to salivary bacteria. In this regard, the paper could be strengthened if there would be some data on viability of bacterial strains in the intestinal tract as the authors might have underestimated the contribution of passively translocated DNA (belonging to living or dead bacteria) to the faecal microbiota. In this regard, a few years ago, Korem et al., (2015) published about this using shotgun metagenomic sequencing to calculate the ratio of sequencing coverage between the peak and trough providing a quantitative measure of a species' growth rate. It would be of importance to see if this bioinformatic approach would help solve this question.

We thank the editor for the constructive comment, and for suggesting the Korem et al., PTR method for growth rates estimation from metagenomic data. To our knowledge, there are currently four available tools that estimate microbial growth rates from sequencing data, two of them published very recently (Nov 2018): PTR (Korem et al., 2015), iRep (Brown et al., 2016, GRiD (Emiola and Oh, 2018) and DEMIC (Gao and Li, 2018). We evaluated all four of these tools in response to the editor’s suggestion.

The original PTR method (Korem et al) requires high quality, closed (i.e., finished) genomes to run; in practice, this requirement is met only by very few genomes even in curated reference databases, as noted by Brown et al., (2016) and Emiola and Oh, (2018). Moreover, the PTR code base has been unmaintained since 2015 and is only available pre-compiled, as noted by Brown et al.:

“As there is no open-source version of the PTR software, we re-implemented the PTR method.”

We did not manage to set up the PTR method to run on our computers. iRep (Brown et al.,), beyond re-implementing the original PTR algorithm, provides an adapted version that handles fragmented genomes spread into multiple contigs. However, iRep requires minimum average coverages of 5x per genome which is a prohibitive requirement when investigating less abundant taxa, as are orally sourced transmitted species in stool samples. We set up iRep on our computers but encountered various run time errors (even on the tutorial data provided by the authors). GRiD (Emiola and Oh) addresses the high coverage requirement of iRep and runs on contigs with as low as 0.2x coverage. We likewise set up this tool, but likewise encountered runtime errors, both on the test data provided by the authors and on real data. For both GRiD and iRep, we contacted the authors about this, but were so far unable to resolve the issues.

DEMIC (Gao and Li) traces coverage of genomes across multiple samples to infer putative oriC and ter sites (this distinguishes the tool from the other three tested). We ran DEMIC on all paired stool samples in our dataset, for all 310 tested taxa. The tool successfully predicted growth rates for 21 species, including four with strong oral-gut transmission signals and one ‘occasional’ transmitter, albeit only in a subset of samples (again, due to coverage cutoffs enforced by the algorithm). For these taxa, we detected active growth, as inferred by peak-to-trough ratios >1, with 23 out of 26 data points (88%) observed in individuals with positive oral-fecal transmission scores. The results are shown below.

This indicates that transmitted taxa grow actively in the gut at least in those samples where inferences were possible. We also added further theoretical arguments why the observed oral-fecal transmission signal cannot be explained by ‘passive’ transmission alone. As detailed in the response to reviewer #3’s original comment below, we now included additional references that free DNA, as released from dead bacterial cells, has a short half-life in the digestive tract, so that we do not expect fecal metagenomic signals to be skewed towards dead oral bacteria after passage through the GI tract.

Finally, a dedicated section on statistical analysis in the method description of the paper would be helpful and the numbers of included individuals should be cross checked within the supplemental data and tables (e.g. Figure 1).

We have now extended the subsection”Diversity, Community Composition and Statistical Analyses” to include more detailed explanations of the individual tests that were performed. Moreover, we have clarified the numbers of samples and individuals in that Figure (now promoted to main Figure 1) and across the main text (see detailed response to reviewer #3’s original comment).

Please see the full reviews below for further points:Reviewer #1:This is a solid paper which shows that bacterial strain exchange between the oral and gut environments is more extensive than previously thought and a normal occurrence, rather than solely a mark of dysbiosis / disease. Nonetheless, they also show that opportunistic pathogens had higher evidence of transmission. While the authors find no correlation between the β-diversity measures (as determined by metagenomic profiling) of the gut and the oral environment, they show that there is an extensive exchange of strains between the two. This is done by determining the strain profiles (SNV profiles of each strain) and showing that, based on probabilistic models, the overlap in some of these profiles is significantly higher than would be expected by random chance ("transmission scores"). The fact that species transmission scores correlate with oral relative abundance, but not to gut abundance, indicate that (as would be expected) the direction of transmission is from the oral cavity to the gut as does the fact that 'oral SNVs observed at an initial time point were significantly enriched among fecal SNVs that were newly gained over time, but generally not vice versa '. The paper is made stronger by the use of longitudinal data.The paper is concise and well-written, providing a detailed, clear and understandable description of the methods (e.g. how the transmission scores are calculated and what the rationale is). The visualizations (including the Supplementary figures) are very expressive and informative. Supporting information files have been submitted, and code and data have been made available on a GitHub repository.I have very few critical remarks to make, I think the paper is rigorous and well-polished:

We thank the reviewer for their encouraging remarks, and for supporting our manuscript.

1) PRJEB28422 accession number does not exist on ENA (checked on Nov 27th). Did the authors mean PRJEB22368?

The project ID given in the manuscript was indeed correct, and data for the corresponding sub-cohorts (DE-CTR and FR-CRC) had been uploaded to ENA under this accession, but the project had (erroneously) not yet been switched to ‘public’. This is now fixed (https://www.ebi.ac.uk/ena/data/view/PRJEB28422); thank you for spotting this.

2) "Transmission scores were negatively correlated with genome size (ρ Spearman=-0.6), indicating that transmitted species generally had smaller genomes than non-transmitted ones" Any idea why this is the case?

The genome size signal stood out among the tested co-variates, though it could in large part be explained by phylogeny (both genome size and transmission scores had a strong, and largely shared, phylogenetic signal). We chose to report this observation, but to abstain from speculation on this point, as true mechanistic hypotheses as to why small genomes are associated with transmission would need to be tested against functional complements, the inference of which is very noisy and incomplete from our data due to the heterogeneous coverage of taxa across oral and fecal samples (see also response to a comment by reviewer #2 below).

3) "the fecal relative abundance of Fusobacterium sp. positively correlated with higher levels of transmission)" Why would this be the case?

We agree with the reviewer that this is an intriguing finding, and we followed this up by building predictive models of oral and gut taxa that were able to classify individuals into ‘high’ and ‘low’ transmission groups with surprising accuracy (new Supplementary file 4 and Figure 3—figure supplement 1). *Fusobacterium nucleatum subsp.* stood out as strong transmission markers. Fn has repeatedly been hypothesized to translocate from the oral cavity to the gut, where its enrichment is strongly associated to colorectal cancer and other diseases. We have now extended the corresponding paragraph to discuss this in more detail.

Reviewer #2:In this work, Schmidt et al., provide theoretical evidence for the transmission and colonization of oral microbial genomes in the distal gut. The results are interesting and important because they show that transmission of oral microbes to the gut occurs extensively in healthy individuals also in adulthood. Therefore, if these results are confirmed, oral-gut transmission might be an important factor to consider for the prevention and management of human diseases through the GIT microbiota.

We thank the reviewer for their constructive and informed comments on our study.

We have a few major concerns that the authors should address to support their findings:1) How were the cut-offs for vertical and horizontal genome coverage chosen? A 5% breadth of coverage seems low for the identification of microbial strains. Would the choice of these cut-offs affect the results?

We filtered our initial species list by a combination of three criteria (>5% horizontal coverage, >0.25x average vertical coverage, >10^-6 relative abundance), with each criterion required to be met in at least 10% of samples (oral and fecal combined). In response to the reviewer’s comment, we have now included the full tables of relative abundance, horizontal and vertical coverages as part of Supplementary file 2, and the number of taxa meeting the depth and breadth criteria alone as Figure 1—figure supplement 5.

The breadth cutoff was chosen to exclude spurious taxa, as in our experience, erroneous mappings to a genome are marked by high (local) depth, but very low (<<5%) breadth; the cutoff value of 5% was chosen based on the plot below that shows the number of included taxa as a function of the breadth inclusion criterion alone. At the same time, we strove to retain an accurate community representation: by applying the above criteria, we removed 1,443 species from the original ‘raw’ set, but these corresponded to only ~1% each of classifiable oral and fecal microbial abundance. Moreover, the above criteria were only applied as an initial filter to define a set of relevant taxa. For each intra-individual and inter-individual SNV-based test, we applied additional cutoffs, as detailed in the Materials and methods section.

Indeed, Figure S4 shows that both vertical and horizontal genome coverage can affect the transmission score at least of some specific taxa. Please highlight the distribution of transmitters in this supplementary figure.

Following the reviewer’s suggestion, we have now highlighted transmitting, occasionally transmitting and non-transmitting taxa in the revised Figure 2—figure supplement 2, and indeed, we observed differences between these groups. Generally, correlations for transmitting taxa were distributed around (or close to) 0, whereas the outlying negative correlations can be ascribed to non-transmitters. This means that for non-transmitters, the “negative” signal is more pronounced the more data is available (deeper coverage, more shared observed genomic positions between the oral and fecal samples), whereas coverage does not generally correlate with transmission scores for transmitting taxa. We feel that this indeed strengthens our original argument that transmission scores (for transmitting taxa) are largely independent of technical covariates, and we thank the reviewer for their suggestion.

2) The authors base their work on the assumption that oral and gut SNVs profiles of transmitted genomes are more similar in an individual than between individuals. However, does this similarity of SNVs profiles necessarily imply transmission? Or, alternatively, could other individual-specific genetic and/or environmental factors shape a similar oral and gut microbiota in an individual instead of transmission?

We agree that while the observed intra-individual SNV overlap is strongly indicative of oral-fecal transmission, other factors could contribute to this signal as well, at least in principle. As we discuss in the text, these could include one-time GI tract-wide colonization events (e.g. during infancy or following a perturbation such as an antibiotics treatment) with subsequent independent evolution at both sites. However, in our opinion, the observed longitudinal signals provide evidence that oral-fecal transmission is an ongoing process even in (healthy) adults: SNVs observed in saliva at t0 are predictive of SNVs gained in feces over time, but not vice versa, and oral-to-fecal transmission rates significantly exceed background expectations.

3) As the authors use a low breadth of coverage and assume transmission based on similarity of SNVs profiles, we would like to ask that the authors confirm their results when using the ConStrains method to identify microbial strains (Luo et al., 2015). This method is also based on SNPs in oral and gut samples.

We fully agree with the reviewer that it would be desirable to corroborate our SNV-based results using an independent strain calling tool such as ConStrains. However, as detailed in our response to the editorial comment above, ConStrains did not detect any strain heterogeneity for any species in any of our tested samples, in line with previously reported tool behavior.

More generally, we had evaluated different strain calling methods at the outset of our study but found that several aspects of our specific research question rendered it outside the space of problems solved by existing tools. In particular, species present in both saliva and stool are usually abundant in one or the other, but almost never at both sites (generally oral species are present at lower abundance in the gut), thereby violating requirements of common tools (e.g., ConStrains requires an average coverage >10x for its inferences). We therefore did not aim to reconstruct comprehensive (genome-wide) strain haplotypes for each individual, which could be useful for some downstream analyses, but are not necessary to infer strain overlap. For this task, observed SNVs, weighted by their cohort-wide background frequencies, can serve as proxies for strain populations, as reported previously (e.g. by Li et al., 2016). In summary, while we did not reconstruct the entire strain space across our samples due to insufficient and heterogenous coverage, we inferred strain overlap between samples within species based on observed marker SNVs.

Reviewer #3:This study entitled "Extensive Transmission of Microbes along the Gastrointestinal Tract" is an original work focusing on the transmission of bacterial strains between the oral and gut environment. The study is a robust analysis at strain-level of the oral and gut microbiota composition intra- and inter-individuals. The data are well exploited, especially regarding potential confounding factors between the different cohorts. The authors set out to identify population flow of bacteria from the oral cavity to the lumen. They defined metagenomes to an SNV-level resolution. The central thesis of this paper is that intra-individual overlap of SNVs between the oral cavity and the lumen is greater than that which would be expected from inter individual background thereby demonstrating oral taxa translocation. Within their model this event of translocation was a persistent one. The taxa identified as transmitted were phylogenetically diverse, yet some clade clustering was noted. The only noted characteristics of these taxa were reduced relative genome size and their anaerobic/ facultative aerobic nature.This is a novel study with potentially significant ramifications for human intestinal microbial ecology.

We thank the reviewer for their encouraging and constructive comments.

My main critiques are these four points:1) One of the pillar arguments in this paper is that the numbers of bacteria cells in the colon which show evidence of transmission, cannot be attributed to the passive translocation due to peristalsis. They argue that the amount of bacterial cells swallowed by a human per day (1.5*1012) would be depleted during passage through the upper digestive tract (stomach and duodenum) and thus would not contribute significantly to the gut microbiota. However, methodologically, the authors are investigating DNA not viable living cells. One might argue the authors have underestimated the contribution of passively translocated DNA (belonging to living or dead bacteria) to the faecal microbiota. The stomach and the mouth contain a high proportion of dead cells (perhaps only 1% of stomach bacteria cells are alive) whose DNA could translocate to the lumen. Given the estimates of an average person has 1 bowel moments a day (the cohort is older with disease so could be less), the average mass of stool to be 100 grams and the bacterial density in stool is estimated to be 0.9·1011 bacteria/g; an individual would pass 9x1012 bacterial cells a day. Presuming that an individual passes all the saliva they swallow a day in their bowel movement and there is no loss of bacterial DNA; One would detect ~1.5*1012 per stool. This would be above the 10% limit that the authors set. I think the authors should further expand on their estimations of quantities of bacterial cells in the lumen that can be attributed to salivary bacteria. The literature to which they reference in not primary work in essence and review. More solid numbers are needed when discussing the oral bacterial density and volume. Indeed, the focus of the Sender et al., 2016 paper is the colonic microbiota. I think their strong claims need stronger support.

The reviewer is right: a central argument of our paper is that ingested salivary bacteria would only be present at abundances below metagenomic detection after passage through the gastrointestinal tract. We realize that one important point regarding this was only made implicitly: the half-life of free DNA (released from dead bacterial cells) in the digestive tract is very short, due to the action of nucleases, the resorption of nucleosides and (to a much lesser extent) uptake of fragments by competent bacteria. Therefore, as only 1 in 100,000-1,000,000 bacterial cells survive passage through the stomach, we expect this reduction to be mirrored in a corresponding decrease in levels of their (intact) DNA after passage of the GI tract, although we recognize that damaged microbial cells are prevalent in feces (see e.g. the recent preprint by Perras et al., 2018). Following the reviewer’s suggestion, we have now referenced studies that quantified DNA degradation in saliva, in the stomach, and in the lower intestine.

Moreover, as per the editor’s suggestion, we have now confirmed that at least some transmitting taxa show indications of active growth as inferred by peak-to-trough ratios >1 in fecal samples (see detailed comment above).

Likewise, the results presented in the study are not enough to support the following statement in particular "Approximately one in three salivary microbial cells colonise in the gut, accounting for at least 2% of the classifiable microbial abundance in faeces".

We thank the reviewer for pointing this out. The phrasing of that sentence is now more precise, referring (more correctly) to the fraction of classifiable salivary microbial cells. The sentence now reads, “Approximately one in three classifiable salivary microbial cells colonize in the gut, accounting for at least 2% of the classifiable microbial abundance in feces.”

2) The number of individuals reported in the text does not match the cohort and dataset overview presented in Figure 1. In the abstract and the main text, the authors reported the analysis of 470 healthy and diseased individuals but based on Figure 1 all together the cohorts comprised 571 individuals including 365 intra-individual couples. Further, the authors reported in the main text they they focussed on a subset of 57 individuals for whom longitudinal data was available but based on Figure 1 only 46 individuals (including diseased individuals) presented time series. Then, for the case-control studies, the authors reported in the main text a total of 172 individuals but based on Figure 1 the cohorts CN-RA, FR-CRC and LU-T1D comprised 395 individuals including 219 intra-individual couples (healthy and diseased individuals). As a general comment, the authors should also clarify precisely in the main text whether the studied individuals are intra-individual couples (with both saliva and stool samples) or individuals with one sample type.

We realize that our terminology in that Figure (now revised main Figure 1) and the main text was not sufficiently precise. We thank the reviewer for pointing this out and apologize for the confusion caused. We counted ‘intra-individual sample couples’ as any paired sampling event for an individual at the same timepoint; for longitudinal cohorts (LU-T1D, CN-RA and DE-CTR), we therefore listed more intra-individual couples than individuals, as some subjects were sampled multiple times. For example, in DE-CTR, five subjects were sampled in one time series each, and provided both saliva and stool at each timepoint (providing 10 intra-individual couples). As detailed in the Materials and methods section, intra-individual longitudinal couples were blocked for when calculating score backgrounds, but we considered each intra-individual saliva-stool couple at each timepoint as a data point.

The previous number of 57 subjects with timeseries was indeed incorrect, as this included CN-RA individuals with fecal-only longitudinal sampling that were not included in the relevant tests. We have now replaced this number by the correct one of 46 individuals.

We have adapted Figure 1 to list both the number of sampling events and the number of subjects. Moreover, following the reviewer’s suggestion, we have now revised the main text and figure caption to clarify this point.

3) The authors profiled 310 prevalent species, which accounted for 99% of classifiable microbial abundance in both saliva and stool. However, there is no mention of the unclassifiable fraction of the reads, the proportion of classified over non-classified reads or the percentage of mapped reads.

In response to this comment, we have now included the full taxa relative abundance table (all 310 species across all tested samples) as part of Supplementary file 2. All relative abundances were scaled to the total number of reads mapping to informative marker genes, and the total classified abundance is reported.

4) The authors should acknowledge the difference between colonization of the lumen (faecal matter) versus colonization of the mucosa. Recent work by Zmora et al., 2018 on probiotics have highlighted the disparity between the colonization of the faecal matter versus the mucosa. I recognize they prepared the current submission before the Zmora paper came out. However, there are many other papers that make this point, at least with respect to faecal versus mucosa.

We thank the reviewer for raising this very relevant point. We have now added a statement to the discussion to clarify that by using feces as a readout, we may indeed underestimate the true extent of oral colonization in the gut, as bacteria from the mucosal linings throughout the GI tract may be under-represented.

[Editors' note: further revisions were requested prior to acceptance, as described below.]

The reviewers were all satisfied with your comments, but one question remains from reviewer 2, on the relative correlations between transmission scores for saliva and gut (see below).Reviewer #2:We thank the authors for their work while revising their manuscript. The manuscript reads well and we have only one additional comment concerning the correlations between average transmission scores and other parameters provided in the supplementary table.We could reproduce the results (Results and Discussion section) for the correlations between transmission scores and prevalence_saliva (rho=0.6) as well as prevalence_gut (rho=0.05). However, there was a stronger correlation between average transmission score and prevalence_gut when accessing only the transmitters.Additionally, strong correlations were observed for average transmission score and P/horizontal coverage for transmitters.Can the authors comment?

We thank the reviewer for their diligent re-analysis of this data subset, and for their valuable comment. We completely agree with the reviewer’s finding and had indeed made the same observations during our own analysis of the data. We now realize that the phrasing in presenting these results was not sufficiently precise.

We tested correlations between transmission scores and ‘technical’ parameters in two ways: as averages of correlations per taxon (presented in Figure 2—figure supplement 2), and as correlations of averages (the results pointed out by the reviewer, Results and Discussion section). The former tests for truly ‘technical’ effects, by correlating transmission scores with co-variates across all samples within a taxon and then testing for systematic associations across all taxa. For example, for *V. parvula* (one of the strongest oral-fecal transmitters in our study), the correlations between transmission scores and horizontal coverage in saliva (0.08) and stool (0.12), vertical coverage in saliva (0.09) and stool (0.01), and salivary abundance (-0.08) across all samples were negligible. When viewed across all taxa (Figure 2—figure supplement 2), and following the reviewer’s previous comment, we concluded that in general, these technical parameters were not positively correlated to transmission scores (in particular for transmitting taxa).

The reviewer’s above findings point to the second type of tests – correlations of averages. For this, we aimed to compare average transmission scores for each taxon across samples to averaged co-variates (prevalence, abundance, coverage). Indeed, fecal prevalence did not globally correlate to average transmission scores (rho=0.05 as reported), but there is a trend for transmitting taxa only (rho=0.67), and similarly for horizontal and vertical fecal coverage (implying abundance). In our view, these are biological rather than technical observations: taxa that are (on average) stronger oral-fecal transmitters are (on average) more prevalent across subjects, and more abundant (covered) in the gut. However, the above tests (Figure 2—figure supplement 2) show that for each individual taxon, transmission scores across subjects are not driven by technical co-variates.

We have now adapted the phrasing in this discussion in the main text and caption of Figure 2—figure supplement 2 accordingly.